# Pharmacoeconomic Analysis of Drugs Used in the Treatment of Pneumonia in Paediatric Population in a Tertiary Care Hospital in India—A Cost-of-Illness Study

**DOI:** 10.3390/medsci5040033

**Published:** 2017-12-11

**Authors:** Lekha Saha, Sharonjeet Kaur, Pratibha Khosla, Sweta Kumari, Alka Rani

**Affiliations:** 1Department of Pharmacology, Post Graduate Institute of Medical Education & Research (PGIMER), Chandigarh 160012, India; sharon25sept@yahoo.co.in (S.K.); pratibha13131@gmail.com (P.K.); sweta6kumari@gmail.com (S.K.); 2Department of Pediatric Medicine, Post Graduate Institute of Medical Education & Research (PGIMER), Chandigarh 160012, India; alkakhadwal@hotmail.com or khadwal.alkarani@pgimer.edu.in

**Keywords:** pneumonia, cost of illness, Pediatric population, developing country

## Abstract

*Aims and objectives:* The cost of antibiotic therapy for the treatment of pneumonia in the inpatient paediatric population can have a major impact on the healthcare expenditure. We planned to assess the direct and indirect costs of diagnosis and medical treatment of paediatric patients with community acquired pneumonia who are hospitalized in a tertiary care hospital in India. *Methods:* 125 children with a diagnosis of pneumonia who were admitted to the inpatient department of a paediatric hospital receiving antibiotic treatment were observed. Data on clinical presentation and resources consumed were collected and the costs of pneumonia treatment were calculated. Descriptive statistics (mean ± standard deviation (SD)) were used to evaluate data regarding demographics, drugs prescribed and cost (direct and indirect cost). Multivariate regression analysis was used to find out predictors of direct and indirect cost. *Results:* Among all pneumonia admissions, mild-to-moderate pneumonia constitutes 76.8%, and 23.2% children were admitted with severe pneumonia; 105 children out of 125 (84%) were suffering from associated disorders along with pneumonia. The majority of antibiotics prescribed belonged to beta lactams (52%) followed by aminoglycosides (19%), macrolides (13%) and peptides (11%). Parenteral routes of administration were used in a majority of patients as compared to oral. The average cost per patient in management of pneumonia was 12245 ± 593 INR ($187.34 ± 9.07).

## 1. Introduction

The global health expenditure database (GHED) was established in 1995 and is probably one of the most comprehensive single data sources for international comparison of the total health expenditure (THE) between various developed and developing countries who are the members of the World Health Organization (WHO) [1]. In 1995, when the GHED was established, the THE composition of the BRICs (Brazil, Russia, India, China) was dominated by Brazil (31%), for China it was 29%, and for India and Russia, it was 20% each [2]. But the recent release of 2012 data shows the entirely different trend, the THE composition of BRICs is now dominated by China (52%), followed by Brazil (17%), Russia (16%) and India (15%) [2]. If we consider the percentage of gross domestic product (GDP) spent on health care systems by these four BRIC countries, it is easy to notice that only India remains at 4% [2]. India’s health expenditure did not match overall economic growth and fell to slightly less than 4% of GDP. In spite of BRICS’ diversity, all countries were able to significantly increase their investments in health care. The major setback was a bold rise in out-of-pocket spending. Most of the BRICS’ growing share of global medical spending was heavily attributable to the overachievement of the People’s Republic of China [3]. So, in a country like India, quantifying the cost of treatment of any disease will help in proper health planning, resource allocation and consumption of limited resources. One of the important causes of morbidity and mortality, which contribute to the excessive health care resource consumption, is respiratory diseases in paediatric populations in both the developed and developing countries. As per the definition, community acquired pneumonia (CAP) is the infection of the lung acquired outside the hospital setting and leads to inflammation of the lung tissue. The main symptoms associated with CAP are fever, cough and tachypnoea, but it may be non-specific in young children. Chest radiography may be useful to confirm the diagnosis. One of the leading causes of death in children throughout the world, especially in developing countries, is CAP. Though the number of under-5 deaths decreased globally from 16.4 million in 1970 to 5.0 million in 2016, there is still a high incidence of CAP in this age group [4]. In 2015, CAP accounted for 15% of deaths in children under 5 years old globally and 922,000 deaths globally in children of all ages [5]. CAP is more common in the developing world, estimated at 0.28 episodes per child per year and accounting for 95% of all cases [6]. According to the World Health Organization data, an acute respiratory illness remains a leading cause of childhood mortality, causing an estimated 1.6–2.2 million deaths globally in children <5 years [4]. The treatment of CAP is still demanding, and outcomes are not predictable, despite the development of newer antibiotics [6]. Therefore, CAP remains an ongoing challenge for the health care facilities and it is a common cause of hospitalization and death in both developing and developed countries [7]. 

The antimicrobial agents used for the treatment of CAP are cephalosporin and penicillin derivatives, macrolides/azalide, newer tetracyclines and respiratory fluoroquinolones [8].

The selection of an antimicrobial agent depends on various factors like: susceptibility pattern of the isolated microorganism, compliance of the patient, the resistance pattern of the organism and adverse event profile of the antimicrobial agent. The patterns of usage of antimicrobial agents differ among the developing and developed countries, thus affecting health care resources. In healthcare facilities, drug usage pattern evaluates prescribing, dispensing, administering and taking of medication and pharmacoeconomic studies offer an opportunity to evaluate the cost of illness [9]. Studies in the literatures have documented the high financial burden of treatment of CAP from the health sector perspective [10,11]. There is not only a burden on the health care system, but there is also a burden on the rest of society, as there is lost productivity for parents and, there is a reduction in quality-of-life for both the children with CAP and their parents.

A loss of an average of 4.2 working days for mothers of a child hospitalized with CAP has been reported in the literatures and there are also other considerable private expenditures [12]. So, there is substantial burden of CAP on society, which, when combined with limited health budgets and the current economic conditions, it is essential to look for alternative treatment strategies to mitigate some of this problem. Studies have been conducted on different treatment approaches of CAP, especially on the efficacy of short-course antibiotics [13,14].

Quantifying cost helps in proper health planning, resource allocation and consumption of limited resources. It is essential due to the fact that financial resources are limited in developing countries and an organizational need generally exceeds available resources. In developing countries like India, there is less or no pharmacoeconomic data on CAP therapy, especially for indirect costs. A survey by Roy et al. was conducted to assess the costs of prescribed medicines and treatment of CAP in Delhi, India [15]. The findings of the study show that the costs of medicines were highly variable and not affordable for the economically poor in India. Modifications in the National Pharmaceutical Policy need to be done urgently. Our aim was to assess the direct and indirect costs of diagnosis and medical treatment of paediatric patients with CAP who are hospitalized in a tertiary care hospital in India.

## 2. Patients and Methods

This was a prospective non-interventional clinical study with retrospective insight into pneumonia-related resource use and the direct costs of medical care, as well as indirect costs associated with absenteeism and the related productivity losses. The study was conducted in the department of Pediatrics in the Post Graduate Institute of Medical Education & Research (PGIMER), Chandigarh, India, from March 2011 to April 2012, with a time zone of one year. The ethical approval was taken from Ethics Review Committee of the PGIMER, Chandigarh, India, before the start of the study (Reference No. MS/1157/Res/249 dated 16 April 2010).

### 2.1. Study Population

One hundred and twenty-five children diagnosed with pneumonia and admitted to the pediatric ward of the PGIMER, Chandigarh, India, from March 2011 to April 2012, were included in the present study. The diagnosis of pneumonia was made based on the following criteria: presence of respiratory symptoms or signs, temperature of ≥37.5 °C or history of fever at home and radiological diagnosis of pneumonia. Patients suffering from community acquired pneumonia were included. Those patients whose primary cause of hospitalization was CAP were only included in the present study. Patients were enrolled consecutively over a period of 12 months. They were assessed by the attending physician (Dr. Alka Rani). There are approximately 200 hospital admissions for CAP, as well as up to 500 cases of CAP treated on an outpatient basis. Approximately 35% of the inpatients initially included were lost follow-up, due to loss of contact. The data which were recorded for each individual child are as follows: clinical presentation, resources consumed like drugs, laboratory tests, length of stay. Regarding drug prescription data, the type of antibiotic prescribed and the dosing scheduled, and duration of antibiotic uses were recorded. As all the children in the present study were less than 15 years of age, informed consent was taken from the parents of the children before they participated in the study. Regarding the collection of economic data like hospitalization cost, cost of drug, costs of various tests and various indirect costs like costs of food, costs of stay, travelling costs, telephone bill, etc. were obtained by interviewing the parents or the guardians.

### 2.2. Outcome Measures

The clinical data were collected from the patient’s case sheets, which included demographic characteristics, diagnosis, length of stay (LOS), drugs used in the management of pneumonia, and other associated information. We included data of admitted patients who had a history of complete recovery from pneumonia. Drug prescription data consisted of type of antibiotic used, dosage, frequency of administration, duration, route of administration and cost.

#### Economic Outcomes

Resource consumption data included costs (direct costs) of different types of laboratory tests, diagnostic tests and procedures, costs of hospitalization, drug costs. Other indirect costs included costs of food, costs of stay, travelling costs, telephone bill and salary curtailed of the parents or a relative of the child during the hospital stay and any other associated costs.

The antibiotic treatment cost was calculated by multiplying the cost of each antibiotic dosage unit by the administered daily dose by the number of treatment days. The cost of the antibiotic dose unit was taken from CIMS (Current Index of Medical Specialties, India) or MIMS (Monthly Index of Medical Specialties, India).

The costs of different types of laboratory tests, diagnostic tests and procedures, costs of hospitalization were calculated as per hospital standard charges. All costs were calculated in Indian currency. The direct costs included the costs of different types of laboratory tests, diagnostic tests and procedures, costs of hospitalization, drug costs. The indirect costs included the costs of food, costs of stay, travelling costs, telephone bill and salary curtailed of the parents or a relative of the child during the hospital stay and any other associated costs. First the direct and indirect costs of each patient were computed, then the average by person is taken.

### 2.3. Statistics

Descriptive statistics (mean ± standard deviation (SD) were used to evaluate data regarding demographics, drugs prescribed and cost (direct and indirect cost). Multivariate regression analysis was used to find out predictors of direct and indirect cost.

## 3. Results

### 3.1. Demographic Characteristics

The study population consisted of 125 children diagnosed with CAP pneumonia: 59.2% of the children were male, while 32.8% were female. The majority of the children were under 5 years of age (90.4%), followed by 6 to 10 years (9%), and only 3% of the children belonged to the age group of 11 to 15 years (Figure 1). The mean age of the children was 1.86 ± 2.65 years. Among all the pneumonia admissions, mild-to-moderate pneumonia constitutes 76.8%, and 23.2% children were admitted with severe pneumonia (Table 1). One hundred and five (105) children out of 125 (84%) were suffering from associated disorders along with pneumonia (Table 2). Congenital diseases were associated with 17.6% of the children. Among congenital diseases, atrial septal defect(ASD),ventricular septal defect (VSD), cleft palate, Down syndrome, patent dactus arteriosus, congenital lymphedema was implicated in the majority of the cases. Infectious causes included sepsis, disseminated staphylococcus abscess, gastroenteritis, etc., and was observed in 19.2% of the children. Respiratory disorders constituted pleural effusion, pyopneumothorax and emphysema, observed in 16.8% of the children. Other associated disorders were liver disease, renal disease, protein energy malnutrition, neurological disorders, hematological disorders, oncological disorders, diabetes mellitus and cardiovascular disorders. Among the total admissions, 42.4% took place in the months of April to September, and 57.6% of the admissions took place in the months of October to March (Figure 2).

### 3.2. Clinical Outcomes

Mean LOS till complete recovery was 11.2 ± 2.2 days. The different types of antibiotics used in treating pneumonia in children were as follows (Figure 3): majority of antibiotics prescribed belonged to beta lactams (52%), followed by aminoglycosides (19%), macrolides (13%) and peptides (11%). Beta lactams used were cephalosporins (Ceftriaxone, cefotaxime, cefoperazone), penicillins (Ampicillin, amoxicillin, cloxacillin), meropenam and combinations such as pipepracillin-tazobactam and amoxicillin-clavulanic acid. Among aminoglycosides class, antibiotics commonly used were amikacin and gentamicin. Macrolide used commonly was azithromycin and peptide used was vancomycin in the majority of the children.

Parenteral routes (intramuscular or intravenous) of administration were used in the majority of patients as compared to oral route; 50% of beta lactam antibiotics were administered through parenteral route followed by aminoglycosides and peptide antibiotics (Table 3).

### 3.3. Economic Outcomes

Table 4 and Figure 4 show the direct and indirect costs of treating pneumonia in children in a tertiary care hospital in India. The average cost per patient in management of pneumonia was 12,245 ± 593 INR ($187.34 ± 9.07). The mean cost of antibiotics was 3823.602 ± 456 INR ($58.50 ± 6.97) (78.13%) as compared to other drugs which cost 1069.742 ± 561 INR ($16.36 ± 8.58) (21.86%) (Figure 5).

While evaluating the factors affecting the direct cost, it was found that as the duration of the hospital stay was increased, the burden of the cost increased; secondly, the cost of the respective antibiotics, which reported that using expensive antibiotics was contributing to higher direct cost (Table 5). Other factors are age, severity of illness, a system of the body involved, etc.

## 4. Discussion

This is the first type of study that reported both the drug usage pattern and descriptive pharmacoeconomic data in a pediatric pneumonia population in a tertiary care hospital in India. More than 90% of the children were under 5 years of age, which is very similar to data pertaining to WHO, which state that the majority of acute respiratory illnesses are found in children under 5 years of age [6]. These data still hold true and appropriate steps at preventive, educational and awareness level are needed to combat such a high incidence of pneumonia at a younger age. In the present study, the majority of the cases of pneumonia (80%) were associated with other disorders such as infectious disorders, congenital heart disease, other respiratory disorders, protein energy malnutrition, neurological disorders, etc., which points towards the other etiologies implicated in pneumonia and take into consideration the other associated disorders by appropriately managing them while reaching the definite diagnosis of the pneumonia [16].

The winter season (March to October) had shown a maximum upsurge of pneumonia cases. Beta lactam antibiotics were prescribed in the majority of the patients as compared to other class of antibiotics. This points towards the sensitivity of microorganism towards the beta lactam but, on the contrary, there should be vigilance regarding indiscriminate and excessive prescribing to prevent emergence of resistance and patient cost. In 50% of patients, beta lactam was administered by parenteral route as compared to 6.8% of children having received oral beta lactam antibiotics. Since pneumonia was associated with other disorders, parenteral route was expected to be used in the majority of the patients and that increased the total costs of treatment in the present study.

Cost analysis depicted that direct cost (drug therapy, laboratory investigations, and admission and bed charges) accounted for the maximum cost as compared to indirect cost. Among the different drugs, antibiotics had a tremendous burden of cost on the patients and their family. This points towards the need to curtail the cost of antibiotics, which are deemed necessary as a part of empiric therapy, which is started before the report of culture and sensitivity (C/S) is reached. After empiric therapy, depending upon the C/S, again antibiotics are either continued or changed which affects the direct cost.

Apart from this, route of administration also affects the cost of the treatment, which means that intravenous drug administration costs more as compared to oral route of drug administration, hence there is a need to go for switch therapy (intravenous to oral therapy) as soon as possible and when the patient can take the drug orally. Study by Lorgelly [17] et al. has also demonstrated the cost-effectiveness of oral amoxicillin for these children. Therefore, there is an alternative therapeutic option which is a potential to achieve cost savings (for both the health service and society) for children hospitalized with pneumonia if treated with oral antibiotics instead of intravenous (IV) antibiotics, as treatment with oral amoxicillin results in a shorter inpatient stay.

The high costs of antibiotics accounted for the higher direct cost in the present study. Thus, cost-minimization studies should be done so that antibiotics, which are showing similar efficacy and low cost, should be incorporated into the prescription of pneumonia [18]. Duration of stay increased the total cost of treatment and decision regarding how long the patient should stay in hospital should be individualized. The total costs of treating pneumonia in children in a developing country such as India at a tertiary care hospital are approximately 12,245 ± 593 ($187.34 ± 9.07), which is especially high for poor patients in our setting and, on behalf of the patient affordability factor, should be taken into account while treating these patients.

Our hospital is a public-sector institution and there are no costs for the physician visits and negligible charges for the nurses. Not only that, all the costs of investigations and the hospital stay are highly subsidized. Thus, the costs of treating inpatient pneumonia children in our hospital are much lower than the costs of treating such patients, not only in developing countries [17,19] but also in comparison to the private-sector institutions of our country.

Limitations of this study are small sample size and children suffering from an associated disorder, which could create bias in the results, especially the cost factor. Disorders such as congenital heart disease, sepsis further necessitates the use of more quantities and higher antibiotics.

## 5. Conclusions

This study established the role of antibiotics as an important part of a therapeutic regimen in CAP.

Antibiotic cost needs to be subsidized looking into their effectiveness in pneumonia, wide usage and lower socioeconomic strata patients presenting in our tertiary care hospital. There has to be a better amalgamation of the drug utilization and pharmacoeconomic studies with clinical practice so that decisions regarding prescribing drugs can be taken into account based on the current results of these studies.

## Figures and Tables

**Figure 1 medsci-05-00033-f001:**
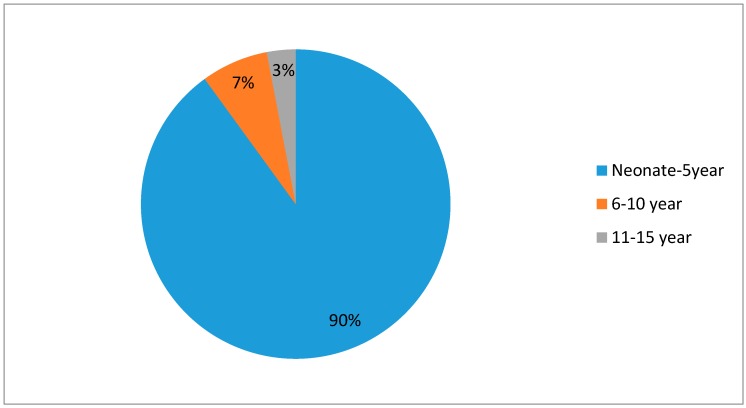
Percentages of children in different age groups.

**Figure 2 medsci-05-00033-f002:**
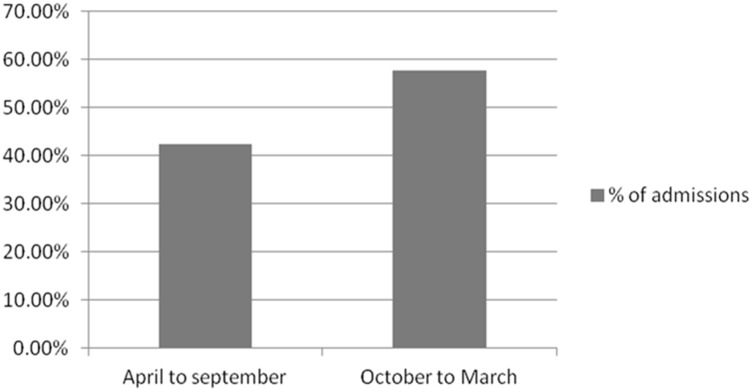
Percentages of admissions in different month.

**Figure 3 medsci-05-00033-f003:**
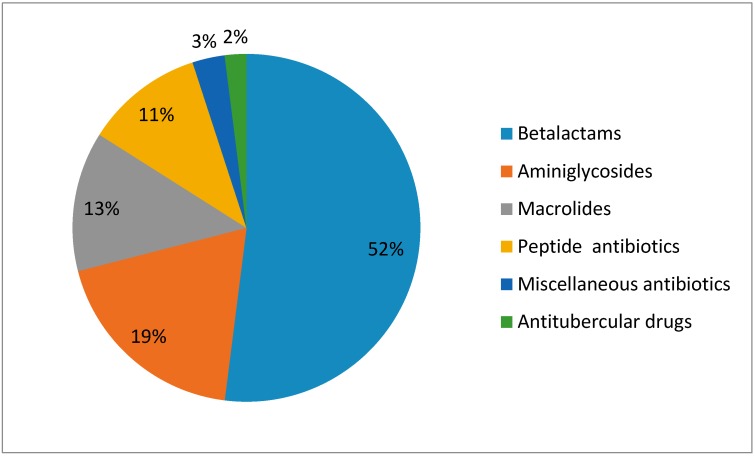
Percentages of prescribed antibiotics.

**Figure 4 medsci-05-00033-f004:**
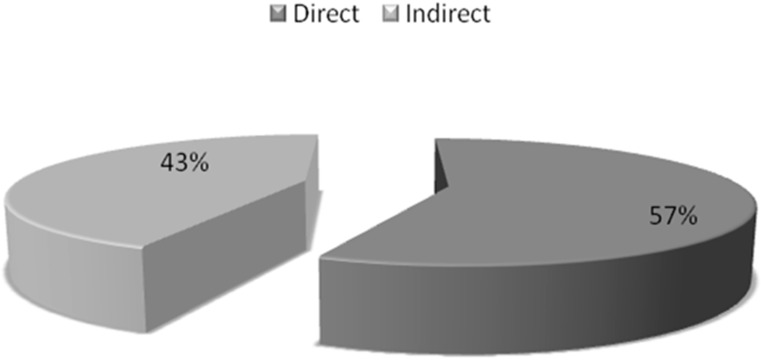
Percentages of direct and indirect costs.

**Figure 5 medsci-05-00033-f005:**
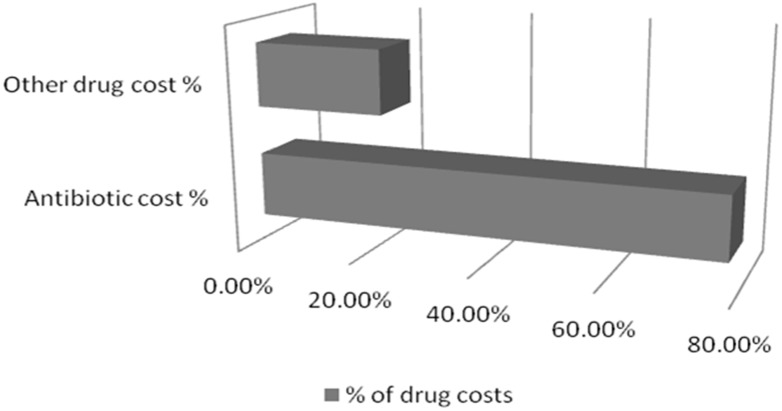
Antibiotic and other drug cost.

**Table 1 medsci-05-00033-t001:** Percentage of children with severity of pneumonia.

Diagnosis	% of Patients
Mild-to-moderate pneumonia	76.8
Severe pneumonia	23.2

**Table 2 medsci-05-00033-t002:** Associated disorders involving different organ systems in children.

No.	System Involved	Percentage (%)
1	Congenital diseases	17.6
2	Liver disease	2.4
3	Renal disease	0.8
4	Protein energy malnutrition	5.6
5	Neurological disorders	4.8
6	Infectious disorders	19.2
7	Hematological disorders	2.4
8	Oncological disorders	2.4
9	Respiratory disorders	16.8
10	Diabetes mellitus	0.8
11	Cardiovascular disorders	4.8
12	Miscellaneous	4.8

**Table 3 medsci-05-00033-t003:** Total encounter with injectable and oral route of administrations.

Antibiotics	Parenteral	Oral
Beta lactams	50%	6.8%
Macrolides	-	3.6%
Aminoglycosides	21.2%	-
Peptide antibiotics	12.04%	-
Antituberculosis antibiotics	-	2.8%
Fluoroquinolones	0.5%	-
Misc.	1.83%	1.04%

**Table 4 medsci-05-00033-t004:** Total costs, direct and indirect costs, and antibiotic costs of treating pneumonia in children.

Category	Parameters	Mean Cost ± SD (INR)	%
Direct Costs	Costs of drug, Investigations costs, Hospitalization costs	6918 ± 104 ($105.84 ± 1.59)	57% of total costs
Indirect Costs	Costs of food, costs of stay, travelling costs, telephone bill and salary curtailed of the parents or relatives	5326 ± 977 ($81.48 ± 14.94)	43% of total costs
Total Costs	Direct Costs + Indirect Costs	12,245 ± 593 ($187.34 ± 9.07)	100%
Antibiotic Costs		3823.602 ± 456 ($58.50 ± 6.97)	78.13% of total drug costs
Other Drug Costs		1069.742 ± 561 ($16.36 ± 8.58)	21.86% of total drug costs
Total Drug Costs		4993.344 ± 467 ($76.39 ± 7.14)	100%

**Table 5 medsci-05-00033-t005:** Coefficients ^a^.

Standardized Coefficients			
Model	Beta	t	Significance
Age	0.055	1.013	0.313
Duration	0.203	3.563	0.001 ^b^
Severity	0.067	1.291	0.199
System	0.074	1.377	0.171
Antibiotics	0.003	0.057	0.955
Cost of antibiotics	0.735	13.675	0.000 ^b^
Sex	0.096	−1.840	0.068
Indirect cost	0.004	0.074	0.941

^a^ Dependent variable: Direct Cost, ^b^ Significant.

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
