# Peer review of "Pharmacoeconomic Analysis of Drugs Used in the Treatment of Pneumonia in Paediatric Population in a Tertiary Care Hospital in India—A Cost-of-Illness Study"

_medsci, 2017, doi:10.3390/medsci5040033_

Reviewer 1 Report

This is a observational study evaluating LOS and costs of pneumonia in a pediatric population of India. The study have a mayor limitation that should be adressed:

The costs should be expressed in US dollars, that will do it more easy to read to scientific around the world. 

Author Response

Q1: English language and style: English language and style are fine/minor spell check required

Response: done

Q2: Are the results clearly presented? Must be improved

Response: Edited the result section and try to improve this section

Q3: The costs should be expressed in US dollars.

Response: done

Reviewer 2 Report

General comments:

I would like to complement the authors on their attempt to estimate the cost of pneumonia-hospitalized episode in a tertiary hospital in India.  The title, cost of illness is misleading. Usually with cost of illness we consider the duration of illness, from the day the patient started to feel unwell to total recovery. In this study, it seems they focused only on the cost associated with a hospitalized episode- this is not a complete cost of illness study.

The paper tried to cover a lot and gave too little details in the methodology. For example, it is not clear if the researchers interviewed patients or their guardians, and if so when did they interview them. I would advise the authors to focus on either the cost or general description of the patients and give more details in methods and results.

Specific comments

1-     Line 47, remove cost, the following sentence imply cost

2-     Line 54: can you give a value for CAP incidence and compare it to other disease to help me understand the magnitude of this problem?

3-     Line 60: remove the repeated “for the”

4-     Line 64: clarify what do you mean by their: Antimicrobial agents or the factors determining their use?

5-     Line 68-69: it is not health care providers, the term is “from the health sector perspective”.

6-     In lines 123 through 125: I don’t think this approach is correct. Since they already have the cost per patient, I am not sure why they did not collect the total direct cost per person, and then add the total indirect cost per person. The computation used is misleading especially if the mother was staying at the hospital and did not need transportation for more than one round trip.

7-     Line 127: It is not clear what cost was included in the indirect cost—if it was for one caregiver or more.

Author Response

General comments:

Q1: English language and style: Moderate English changes required

Response: We have tried our best to improve the English language of the manuscript. Still, if the editor or the reviewer feels further editing of the language, they can do so.

Q2: The title , cost of illness is misleading. Usually with cost of illness,  we consider the duration of illness, from the day the patient started to feel unwell to total recovery. In this study, it seems they focused only on the cost associated with a hospitalized episode. This is not a complete cost of illness study.

Response: Though the reviewer is right, In the present study, we included all the costs related to the diagnostic tests or procedures for the CAP and all the drug costs along with hospitalization costs and all indirect costs like costs of food, costs of stay, travelling costs, telephone bill and salary curtailed of the parents or a  relative of the child during the hospital stay and any other associated costs. The parents or guardians of the child were interviewed regarding the costs associated with CAP treatment before coming to our hospital  and till the child recovered fully from the pneumonia. Still, if the editor or the reviewer feels that we should change the title of the manuscript we can do it.

Q3: The paper tried to cover a lot and gave too little details in the methodology. For example, it is not clear if the researchers interviewed patients or their guardians, and if so when did they interview them. I would advise the authors to focus on either the cost or general description of the patients and give more details in methods and results.

Response:  The results and methodology have been modified.

Specific comments:

1. Line 47, remove cost, the following sentence imply cost

Response: corrected

2.  Line 54: can you give a value for CAP incidence and compare it to other disease to help me understand the magnitude of this problem?

Response: edited this part of introduction

3.  Line 60: remove the repeated “for the”

Response: remove

4. Line 64: clarify what do you mean by their: Antimicrobial agents or the factors determining their use?

Response: clarify in the text.

5. Line 68-69: it is not health care providers, the term is “from the health sector perspective”

Response: Corrected

6. In lines 123 through 125: I don’t think this approach is correct. Since they already have the cost per patient, I am not sure why they did not collect the total direct cost per person, and then add the total indirect cost per person. The computation used is misleading especially if the mother was staying at the hospital and did not need transportation for more than one round trip.

Response: the direct and indirect costs of individual child was first calculated and then do statistics. they transportation cost was calculated once if the mother was staying at the hospital.

7.   Line 127: It is not clear what cost was included in the indirect cost—if it was for one caregiver or more.

Response: The indirect costs include: costs of food, costs of stay, travelling costs, telephone bill and salary curtailed of the parents or a  relative of the child during the hospital stay and any other associated costs. If more than one caregiver were there for a single child, then the total costs of all caregiver was calculated.

Reviewer 3 Report

Dear Authors,

This is a decent health economics contribution on India.

In my opinion it is valuable of publishing as it fills important knowledge gap.

The one major weakness that should be corrected is to extend your current reference list with the appropriate health economics published evidence from high impact journals I have provided within my manuscript revision uploaded.

(yellow marked sentences with marginal comments)

Conditional to adopting majority of these suggestions for improvement I would like to warmly embrace your paper for publishing.

Sincerely,

Peer Reviewer

Author Response

Q1: English language and style: Moderate English changes required

Response: We have tried our best to improve the English language of the manuscript. Still, if the editor or the reviewer feels further editing of the language, they can do so.

Q2: The one major weakness that should be corrected is to extend your current reference list with the appropriate health economics published evidence from high impact journals I have provided within my manuscript revision uploaded.

Response: the introduction has been modified and current references have been cited in the text. All the current references have been incorporated throughout the text.

Round  2

Reviewer 1 Report

Saha et al studied the cost of hospitalizations for pediatric CAP.

The study was improved; however, there are some issues that should be adressed:

- The introduction is so large and discuss irrelevant data about expense in health care of BRICS countries. An important data that should be described in the introduction is the GDP of the region.

- The authors should describe and discuss mainly limitations of the study: 

- The costs are only valid for India, in other countries the cost of hospitalizations are higher. 

- The authors included patients with other infectious diseases (It is not clear what is the primary cause of hospitalizations).

- Why did the authors excludes other cause of respiratory infections?

Author Response

The study was improved; however, there are some issues that should be adressed:

Q1: The introduction is so large and discuss irrelevant data about expense in health care of BRICS countries. An important data that should be described in the introduction is the GDP of the region.

Response:  The introduction has been modified accordingly

Q2: The authors should describe and discuss mainly limitations of the study: 

Response: Mentioned in the discussion

Q3: The costs are only valid for India, in other countries the cost of hospitalizations are higher. 

Response: Mentioned in the discussion

Q4: The authors included patients with other infectious diseases (It is not clear what is the primary cause of hospitalizations).

Response: Those patients whose primary cause of hospitalization was CAP were only included in the present study.

Q5: Why did the authors excludes other cause of respiratory infections?

Response: Because the study was on CAP only.